# Gender Differences in Anxiety, Attitudes, and Fear among Nursing Undergraduates Coping with CPR Training with PPE Kit for COVID

**DOI:** 10.3390/ijerph192315713

**Published:** 2022-11-25

**Authors:** Clara Maestre-Miquel, Francisco Martín-Rodríguez, Carlos Durantez-Fernández, José L. Martín-Conty, Antonio Viñuela, Begoña Polonio-López, Carmen Romo-Barrientos, Juan José Criado-Álvarez, Francisca Torres-Falguera, Rosa Conty-Serrano, Cristina Jorge-Soto, Alicia Mohedano-Moriano

**Affiliations:** 1Department of Nursing, Physiotherapy and Occupational Therapy, Faculty of Health Sciences, University of Castilla-La Mancha, 45600 Talavera de la Reina, Spain; 2Faculty of Medicine, Universidad de Valladolid, 47005 Valladolid, Spain; 3Prehospital Early Warning Scoring-System Investigation Group, 47005 Valladolid, Spain; 4Advanced Life Support, Emergency Medical Services (SACYL), 47007 Valladolid, Spain; 5Department of Nursing, Faculty of Nursing, University of Valladolid, 47002 Valladolid, Spain; 6Technological Innovation Applied to Health Research Group (ITAS), Faculty of Health Sciences, University of Castilla-La Mancha, 45600 Talavera de la Reina, Spain; 7Integrated Attention Management of Talavera de la Reina, Castilla-La Mancha Health Service (SESCAM), 45600 Talavera de la Reina, Spain; 8Faculty of Health Sciences, University of Castilla-La Mancha, 45600 Talavera de la Reina, Spain; 9Institute of Health Sciences of Castilla-La Mancha, 45600 Talavera de la Reina, Spain; 10Faculty of Nursing, University of Castilla-La Mancha, 45071 Toledo, Spain; 11Faculty of Nursing, University of Santiago de Compostela, 15782 Santiago de Compostela, Spain

**Keywords:** anxiety, simulation training, personal protective equipment

## Abstract

Background: The aim of this study was to examine the attitudes, fears, and anxiety level of nursing students faced with a critical clinical simulation (cardiopulmonary reanimation) with and without personal protective equipment (PPE). Methods: A pilot before–after study as conducted from 21 to 25 June 2021, with 24 students registered in the nursing degree of the Faculty of Health Sciences of the Castilla-La Mancha University (UCLM) in the city of Talavera de la Reina (Toledo, Spain). From 520 possible participants, only 24 were selected according to the exclusion and inclusion criteria. The STAI Manual for the State-Trait Anxiety Inventory, a self-evaluation questionnaire, was used to study trait STAI (basal anxiety), trait STAI before CPR, state STAI after CPR, total STAI before CPR, and total STAI after CPR as the main variables. A *t*-test was used to study the STAI variables according to sex and the physiological values related to the anxiety level of participants. An ANOVA statistical test was used to perform a data analysis of the STAI variables. Results: A total of 54.2% of participants (IC 95% 35.1–72.1) suffered from global anxiety before the cardiopulmonary reanimation maneuvers (CPR). The results of the STAI before CPR maneuvers showed significant differences according to gender in state anxiety (*p* = 0.04), with a higher level of anxiety in women (22.38 ± 7.69 vs. 15.82 ± 7.18). Conclusions: This study demonstrates different levels of anxiety in terms of gender suffered by nursing students in high-pressure environments, such as a CPR situation.

## 1. Introduction

Nursing education has frequently been linked to anxiety among undergraduate students [1,2,3]. Anxiety helps prepare an individual to respond and act appropriately to a situation [4,5], which is crucial in health-assistance settings. Anxiety has been detailed previously in other studies [6,7,8], especially among women [9,10].

Furthermore, the clinical training taking place during nursing education is more stressful than other theoretical subjects [1,11] and more stressful than other clinical practices among undergraduates in different health degrees. Particularly, anxiety among nursing students has a negative effect on their quality of life [2] and may cause them to drop out of their programs [3]. It is also well-known that negative emotions and feelings such as fear and anxiety can influence the learning process in nursing students [12,13,14]. High levels of anxiety, fear, and other feelings can make learning difficult [15] and even influence decision making [16,17]. Three important aspects can influence decision making: experience, knowledge, and the emotional, mental, and physical state of the student or nurse [18]. Incorrect decision making can stem from a lack of knowledge and/or learning [19] and even from sociodemographic variables such as sex and age [20].

Due to the COVID pandemic, the scenario for health professionals has only gotten worse [21], particularly for recent nurses and students in practice [22]. On the one hand, age, expertise, and concerns about infection risk increase the risk of suffering anxiety among frontline healthcare workers fighting COVID-19 [23], and on the other hand, a lack of PPE and fear of infection can increase the risk of anxiety among nursing students [21]. There is also a gender association: there is a higher anxiety level in female students than in males [2,24,25,26,27].

Given the fact that females comprise most of the undergraduates in nursing careers worldwide, gender can in part explain the focus on the high prevalence of anxiety studies in many publications [9,10,21].

There are many studies on the prevalence of anxiety during clinical simulations that resemble life situations [22,28], outlining those individuals who were subjected to the pressure of the intervention, the critical state of the patient, and the aspects related to the patient’s death [29,30,31]. With respect to the effect of wearing personal protective equipment on CPR quality during the pandemic, Rauch et al. [32] have recently published the results from a sample of providers from the prehospital emergency medical service. In that study, the authors did not find any effect of wearing PPE with respect to compression depth, release, or rate or number of effective compressions.

However, there is no literature about the impact of wearing a PPE (personal protective equipment) on state or trait anxiety during a clinical simulation.

Since stress and anxiety affect the decision-making and learning process [17], it is necessary to conduct studies that include the factor of wearing PPE during clinical simulations with nursing students in order to enable for them to work in the COVID scenario or for others that require the same level of biosecurity.

Thus, the aim of this study was to examine the attitudes, fears, and anxiety level that nursing students experience when faced with a critical clinical simulation (cardiopulmonary reanimation) with and without personal protective equipment.

## 2. Materials and Methods

### 2.1. Study Design

A pilot before–after study [22] was conducted from 21 to 25 June 2021, with 520 students registered in the nursing degree of the Faculty of Health Sciences of the Castilla-La Mancha University (UCLM) in the city of Talavera de la Reina (Toledo, Spain). Of 520 possible participants, only 73 were selected according to the inclusion criteria. Using random numbers generated by computer software XLSTAT ^®^ BioMED 14.4.0 (Microsoft Inc., Redmond, WA, USA), we appointed a final sample of 24 participants (Figure 1).

Before performing the CPR simulation, all participants had received a CPR seminar with dummies. All selected participants were aged between 18 and 34. They all had basic knowledge of the maneuvers, according to the American Heart Association or European Resuscitation Council training.

### 2.2. Population

The eligible population were nursing students with accredited knowledge in basic cardiopulmonary resuscitation. We performed a random selection of subjects who showed interest in participating in the study and did not present any of the exclusion criteria. The exclusion criteria were similar to those considered in previous studies [28] (major surgery in the last 30 days; blood sugar levels < 65 mg/dL; electrocardiogram with alterations; resting heart rate of >120 beats/minute (bpm) or <35 bpm; body mass index > 40 kg/m^2^; temperature > 38 °C; systolic or diastolic blood pressure > 160 or <95 mmHg or systolic blood pressure < 80 mmHg, respectively; any type of functional disorder hindering cardiopulmonary resuscitation maneuvers; oxygen saturation < 92%; acute-phase skin diseases or systemic immunological diseases; severe visual or hearing impairment or epilepsy; diagnosed infections treated while the study was conducted).

The sample size was 24 subjects, 11 (45.8%) men and 13 (54.2%) women; it was calculated as accepting a 0.05 alpha risk. All participants signed the informed consent and carried out the study.

### 2.3. Study Protocol

In this pilot study, each subject carried out two interventions, with 24 h resting time in between: one intervention without the use of personal protective equipment (PPE) and another with complete PPE (that is, protective equipment for infection control including FFP2 mask, coverall, clothing, doble gloves, protective glasses, gown). To eliminate any compliance bias, all subjects had the same probability of being included in any group in the first intervention, as a randomization sequence was generated using random numbers according to the gender stratification created by the computer with Microsoft Excel^®^ version 14.4.0. (Microsoft Inc., Redmond, WA, USA).

First, all students completed a State Trait Anxiety Inventory (STAI) (State Anxiety-SA and Trait Anxiety-TA) questionnaire and a “Pre-feelings and emotions” questionnaire thirty minutes before the performance of CPR. They continued on to the different scenarios in which they performed a test for 10 min with a high-quality CPR simulator: Real CPR Help software installed in an R Series monitor-defibrillator (ZOLL Medical Corporation, Chelmsford, MA, USA) and CPR-D-padz^®^ defibrillation electrodes (ZOLL Medical Corporation, Chelmsford, MA, USA) (Real CPR Help^®^ technology provides real-time feedback about the depth and frequency of CPR while it is applied, which provides guidance on improving the quality of the CPR).

At the end of the performance, the same State Trait Anxiety Inventory (STAI) and “Post-feelings and emotions” questionnaires were completed.

At no time was it reported that the students had to complete the questionnaires after completing the CPR test to avoid bias.

The State Trait Anxiety Inventory (STAI) is a self-administered questionnaire, validated for the Spanish population, and it has a Cronbach’s alpha of 0.93 for TA and 0.92 for SA [33]. This instrument measures anxiety in healthy adults. It has two scales: state anxiety (SA, reflects temporary anxiety about a particular event) and trait anxiety (TA, reflects anxious propensity that characterizes individuals), with 20 questions each. The questionnaire provides a numerical value for TA and another for SA [28]. The total STAI is the sum of SA and TA.

In addition, “Pre- and Post-feelings and emotions” questionnaires were anonymous, non-validated, and based on the model presented by Miquel Perez et al. [34] and Romo-Barrientos et al. [9]. These instruments were administered to characterize students’ feelings and emotions regarding the CPR maneuvers with/without PPE.

The “Pre-feelings and emotions” questionnaire consisted of 12 questions, and the “Post-feelings and emotions” questionnaire consisted of 16 questions (adding four new questions related to students’ satisfaction and emotional experience performing the CPR) (Appendix A).

### 2.4. Data Analysis

Data were checked for meeting the normality condition with the Shapiro–Wilk test. Categorical variables were described using absolute frequencies with a 95% confidence interval (IC 95%), considering descriptive statistics, means, and standard deviation (SD). In the descriptive and inferential statistical analysis, the parameters were used according to the scale of the variable. The quantitative variables herein contemplated were as follows: trait STAI (basal anxiety), trait STAI before CPR, state STAI after CPR, total STAI before CPR, and total STAI after CPR.

With a *t*-test, we studied the STAI variables according to sex and the physiological values related to the anxiety level of participants. An ANOVA statistical test was used to perform a data analysis of the STAI variables according to the experimental group. A 95% confidence level was established. The SPSS statistical package, v. 24 (SPSS Inc., Chicago, IL, USA) and XLSTAT ^®^ BioMED software were employed.

### 2.5. Ethical Considerations

The participants were informed about the general objectives of the study and gave their informed consent. The study was approved by the Clinical Research Ethics Committee of Talavera de la Reina (Toledo) with number 178013/113. Details of the study design, statistical analysis plan, and baseline data are available online (doi.org/10.1186/ISRCTN10222040 (accessed on 5 October 2022)).

## 3. Results

We included 24 subjects, of whom 54.2% were women and 45.8% were men, with a mean age of 22.12 years (SD 3.84). All participants belonged to the nursing degree according to the academic course. A total of 25% of the sample were students from the first course, 25% were from the second course, and 50% were from the fourth course (Figure 1).

Regarding the STAI questionnaire, in the first phase (PRE), the score for the STAI-TA was 22.46 ± 8.57 points. The STAI-SA decreased from the first phase (19.40 ± 8.03 points) to the second phase (16.04 ± 8.51 points) without statistically significant differences (*p* > 0.05). We found significant differences between STAI-state anxiety among males and females, where female participants showed a higher level of anxiety (*p* = 0.04).

Regarding to the STAI-SA, we found significant differences between the first phase (PRE) and the second phase (POST) in the female group (*p* = 0.016).

When studied with regard to PPE, there were no significant differences (Table 1).

According to Romo-Barrientos et al. [9], Spielberger et al. [33], Arraez-Aybar et al. [35], and Casado et al. [36], we consider that a subject suffers from global anxiety when their trait anxiety minus state anxiety is less than six points (TA-SA > 6). Therefore, in the sample, 54.2% (IC 95% 35.1–72.1) of participants suffered global anxiety before the CPR, and 62.5% (IC 95% 42.7–78.8) suffered global anxiety after the CPR, without significant differences. This means a relative increase of 8.3 points in percentage.

The results obtained from the questionnaire indicate that the main thoughts and feelings regarding the cardiopulmonary reanimation were *uncertainty* and *curiosity*, at 62.5% (n: 15) and 58.3% (n: 14), respectively. These were described with a frequency higher than other feelings, such as *fear* or anxiety.

According to the question “What is your main concern about performing a CPR with PPE?”, the students answered that their greatest concern was *not performing a quality CPR* (75%, n: 18) and *the patient’s death* (45.8%, n: 11) vs. *a possible contagion* (12.5%, n: 3) (Figure 2), without significant differences (*p* > 0.05). When we studied differences according to gender, we found a significant difference for the answer *uncertainty*, with more males than females acknowledging this feeling (*p* = 0.035).

In general, during the first phase (PRE), the students were “calm” (n: 16; 66.7%) and felt “safety” (n: 14; 58.3%) more than “nervous” (n: 13; 41.7%), “worried” (n: 6, 25.0%), or “afraid” (n: 3, 12.5%).

After the intervention (POST), there was an increase in the average of “calm”, “safety”, and “relaxed” (n: 20, 83.0%; n: 16, 69.6%; n: 15, 62.5%, respectively). The students were less “nervous” (n: 3; 65.2%) and felt a slight increase in being “afraid” (16.7%, n: 4) compared with the feelings perceived in the first phase. Nevertheless, 70.8% were satisfied with themselves (n: 17) (Figure 3).

Regarding the comparison among the self-perceived feelings questionnaire PRE- and POST-intervention, there were no statistically significant differences (*p* > 0.05) in any variable except nervousness (*p* < 0.01 before vs. after) (Figure 3).

The perception of the participants towards performing the CPR maneuvers was the main outcome of this study. In the first phase, students were asked about the feelings they experience when performing CPR under biological risk conditions or during a pandemic, in which case the answers were “uncertainty” and “curiosity” (62.50% and 58.30%, respectively) and facing “anxiety” and “dislike” (12.50% and 4.20%, respectively) (Figure 2).

When performing the maneuvers with the PPE kit, the participants worried about *the patient’s death* (27.90%) and *not performing a quality CPR* (75%), whereas a *possible contagion* was only a worry of 12.5% of the sample, considering that 50% of the participants had worn a PPE kit before this study (Figure 2).

Regarding the evaluation of training in the second phase, the students answered that they needed more training to improve their proper technique and concentration to complete a quality CPR (79.2%, n: 19). A percentage of 87.5% (n: 21) considered that it was necessary to carry out more training in CPR throughout the nursing degree, and 87.5% recommended this training activity for other courses (n: 21) (Figure 2).

The score given by the students in this training was 8.62 ± 1.04 points over ten points, and 95.8% (n: 23) were satisfied or very satisfied with the training activity.

## 4. Discussion

This study attempted to examine the attitudes, fears, and anxiety level that nursing students experience when faced with a critical clinical simulation (cardiopulmonary resuscitation) with and without a personal protective equipment (PPE). The literature shows that simulation scenarios facilitate objective teaching models, which would otherwise be limited by ethical, social, administrative, and legal parameters [37]. Several studies have described a lack of confidence, a transfer of experienced emotions to the simulation environment, as well as a lack of evidence compared with traditional methods [38]. However, this is the first time that anxiety, emotions, and attitudes during a critical clinical simulation have been studied according to the context of the COVID pandemic, using a PPE kit.

The STAI questionnaire was found to be a useful tool in the evaluation of state and trait anxiety level. The results of the statistical analysis for the STAI before the CPR maneuvers showed significant differences by gender on state anxiety, with a higher level of anxiety in females before the CPR. This is consistent with studies in which differences between men and women have been observed in critical care residents in CPR simulation scenarios [39]; in extreme conditions, gender differences have been observed after CPR simulations, with a higher level of anxiety in women [28]. During the first wave of the COVID pandemic, an increase in anxiety in female health professionals was evidenced [40,41,42,43,44,45]. Concurrently, other studies with undergraduates have shown higher levels of trait anxiety in women [9,10].

In addition, we found that the female group showed a higher level of anxiety in the first phase of the simulation. These results raise the question as to whether the higher anxiety scores occurred before the CPR maneuvers. What is known is that the measurement of stress is related to a transient event, and that once it ends, anxiety values decrease quickly [46]. These data are consistent with our study. In addition, this prior anxiety can affect learning, decision making, and CPR execution, confirming studies that show that anxiety affects learning and decision making [12,14].

Our results suggest an increase in anxiety at the final phase due to the doubts of the participants about the real outcomes of their performance, as the feeling of being observed or the anticipatory anxiety is diminished, which coincides with recent findings [47].

The outcomes of this pilot study indicate that the main concern about the use of a PPE kit during a cardiopulmonary resuscitation is not only the patient’s death but also the quality of the maneuvers, relegating the fear of a possible contagion to a secondary place. Recently, other factors such as the effectiveness of time, the perception of a safe environment in contrast to the real context, or findings of making a mistake while being observed have been related to stress and anxiety during a CPR [48].

This pilot study also indicates that knowing how to perform CPR does not presume calm, comfort, or an ability to control state anxiety in critical situations. The psychological response of health professionals in emergency situations is a frequently discussed issue. It has been suggested that PPE use is a stressor for all response personnel; other authors suggest that the trait anxiety is most affected by using PPE. Dunbar’s works are the precedents that describe how the stable psychological traits of anxiety and the expression of anger mitigate against performance in using PPE [49].

Nursing education has traditionally been a female discipline [50], and this election depends on the gender roles and social factors [51]. Nursing studies currently have a higher percentage of female undergraduates, surprisingly, with statistical differences in stress by gender with worse results in women [52], which could affect the learning process and academic scores and self-esteem, even the decision-making process [53,54].

Recent studies have examined job stress-coping strategies among nurses. These results could be used to develop specific training in nursing students. Some of the most emotion-oriented coping strategies used by nurses include positive reappraisal [55], which demonstrates higher psychological competences and significantly better professional behavior and personality traits [56]. Positive reappraisal has demonstrated benefits in terms of psychological health [57] and, in general, females are more likely to use positive reappraisal [58]. We agree that specific emotional training when facing to a critical clinical scenario would improve the results in making decisions and also the performance of quality resuscitation maneuvers.

Finally, as strengths, this study has included a specific scenario with a PPE kit for COVID, which involved the complex use of resources and organization, and results are revealed that have not been published previously. Further, other clinical simulation studies have used different questionnaires to measure anxiety, such as Nursing Anxiety and Self-confidence with Clinical Decision-Making (NASC-CDM), although the most frequently used measure in these cases is the STAI, as a validated and effective tool for measuring self-perceived anxiety. In addition, we measured physiological variables that strengthen the results regarding the effects of anxiety. However, this preliminary study limits a generalization of findings owing to the small sample size. More studies with larger sample sizes that also consider additional variables, such as culture, socioeconomic level, and beliefs, are required to provide in-depth understanding.

One of the weaknesses of the study is the sample size, which was based on an opportunity sample, meeting the inclusion and exclusion criteria method.

## 5. Conclusions

This study has demonstrated the different levels of stress and anxiety in terms of gender that nursing students in high-pressure environments perceive, such as a CPR situation. Future stress-management interventions and training should be developed using positive coping strategies to enhance supportive working environments to enable nurses to provide better quality care to critical patients.

## Figures and Tables

**Figure 1 ijerph-19-15713-f001:**
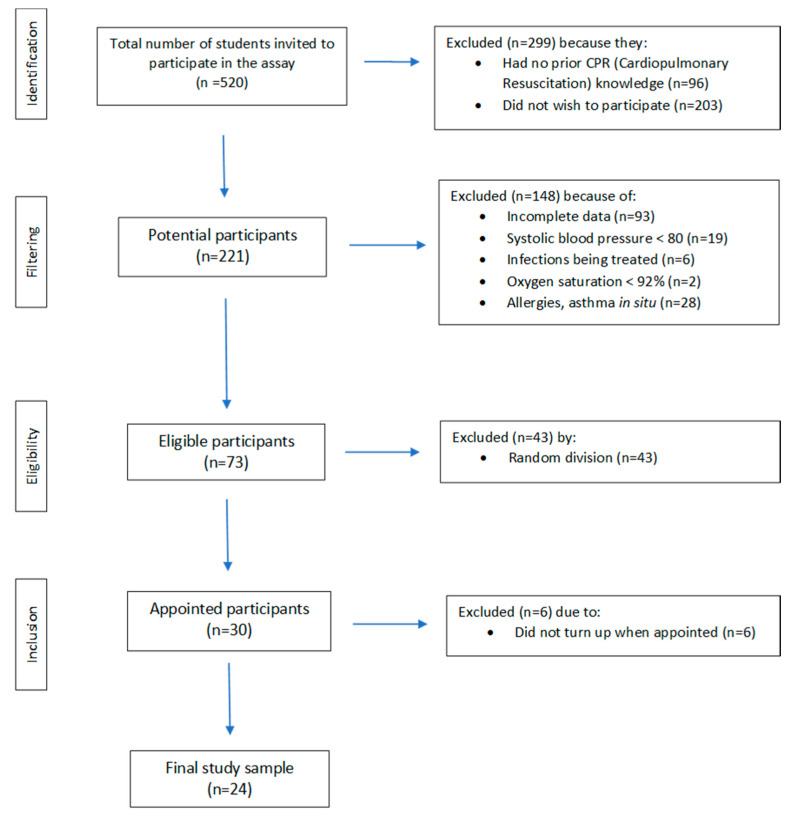
Flow chart of the selection of participants for this study.

**Figure 2 ijerph-19-15713-f002:**
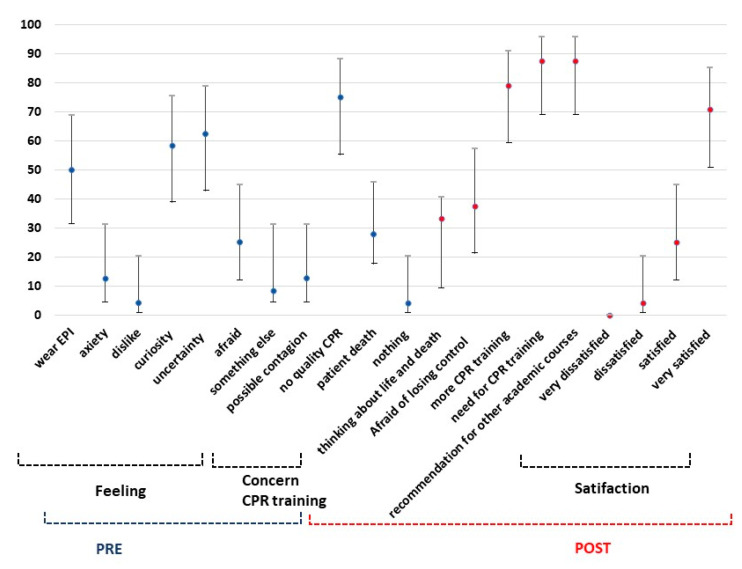
Prevalence of students’ thoughts and feelings (with IC 95%) regarding to the cardiopulmonary reanimation.

**Figure 3 ijerph-19-15713-f003:**
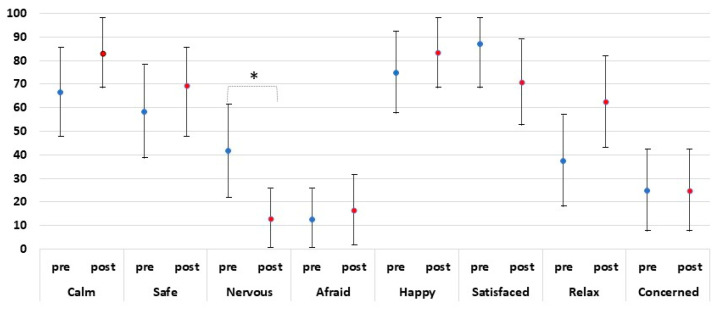
Comparison among self-perceived feelings pre- and post-CPR (IC 95%). * Statistically significant differences (*p* < 0.01 before vs. after) are shown.

**Table 1 ijerph-19-15713-t001:** Anxiety (pre- and post-intervention) categorized by gender and by personal protective equipment use.

	PRE-Trait Anxiety	PRE-State Anxiety	POST-State Anxiety	*p*-Value PRE-Trait Anxiety vs. PRE-State Anxiety vs. POST-State Anxiety
Total	22.46 ± 8.57	19.40 ± 8.03	16.04 ± 8.51	0.088
Men	21.82 ± 8.39	15.82 ± 7.18	17.27 ± 9.36	0.449
Female	23.00 ± 9.02	22.38 ± 7.69	15.00 ± 7.95	0.016
***p*-value** **Men vs. Female**	0.744	0.04	0.526	
IPE	21.15 ± 8.23	20.31 ± 9.18	16.00 ± 10.13	0.176
No IPE	24.00 ± 9.09	18.27 ± 6.66	16.09 ± 6.58	0.331
***p*-value** **IPE vs. No IPE**	0.430	0.548	0.980	

## Data Availability

doi.org/10.1186/ISRCTN10222040 (accessed on 5 October 2022).

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
