# Peer review of "Gender Differences in Anxiety, Attitudes, and Fear among Nursing Undergraduates Coping with CPR Training with PPE Kit for COVID"

_ijerph, 2022, doi:10.3390/ijerph192315713_

Round 1
Reviewer 1 Report
Attitude and fear are not explained in the title, methods and results. The study variables should be consistent mentioned or explained in the title, aim, method, results, and conclusion.
Explain sampling method, data collection, criteria of respondents in the methods of the abstract.
Was stress measured in the study? Why it is mentioned in the conclusion of the abstract.
Explain theoretical based in the study.
Why do authors described about coping strategies in the background? IMHO, it should not be explained since the study is not investigate coping strategies.
Statement in line no. 60-61 about anxiety is not related coping strategies as described in previous sentences.
The study phenomenon regarding CPR training wearing PPE of COVID should be described in the background yet.
The study design (cross-sectional) is not appropriate if the researchers conducted interventions.
How did the researcher calculate the sample size and randomize the participants?
Is it ethical to provide intervention without PPE?
Did the researcher conduct ethical clearance? It is important since student is one of vulnerable subjects in research.
Please describe normality test result as the requirement to use t-test.
Was there any qualitative study conducted in the study because the researchers asking about main concern about performing a CPR with PPE?
Explanations about coping in the discussion are not consistent with results.
Author Response
Dear Editor,
We wish to thank you all for your constructive comments in this round of review. Your comments provided valuable insights to refine our contents and analysis.
Thus, the manuscript has been revised according to the suggestions and comments of the reviewers. Please also kindly note that the title of the manuscript has been changed to “Gender Differences in Anxiety, Attitudes and Fear, among Nursing Undergraduates Coping with CPR Training with PPE Kit for Covid”.
The responses to the specific comments of the reviewers are as follows:
From reviewer 1:
- Attitude and fear are not explained in the title, methods and results. The study variables should be consistent mentioned or explained in the title, aim, method, results, and conclusion.
Answer: we already have changed the title to better understand the main variables studied. We also have reviewed the rest of the manuscript in those terms.
- Explain sampling method, data collection, criteria of respondents in the methods of the abstract. Was stress measured in the study? Why it is mentioned in the conclusion of the abstract.
Answer: according to the reviewer suggestions, we have completed the Abstract and we have rejected the mention of the stress, because it´s do related but it hasn’t been measured on the study.
Line 29: A before–after study was conducted from 21 to 25 June 2021, with 520 students registered in nursing degree in the Faculty of Health Sciences of the Castilla-La Mancha University (UCLM) in the city of Talavera de la Reina (Toledo, Spain). From 520 possible participants, only 24 were selected according to the exclusion and inclusion criteria (…).
Line 41: Conclusions: This study has demonstrated different level of anxiety by gender that nursing students suffer in high-pressure environments, such as a CPR situation.
- Explain theoretical based in the study. Why do authors describe about coping strategies in the background? IMHO, it should not be explained since the study is not investigate coping strategies. Statement in line no. 60-61 about anxiety is not related coping strategies as described in previous sentences.
Answer: Thanking for the comments, we have rejected the sentence in line 59-61, talking about coping strategies in the Introduction, and we have added a wider vision of the current state of this issue, adding a recent reference about it.
Line 62: Due to the COVID pandemic, the scenario for health professionals has only got worse [21], particularly for recent nurses and students in practice [22]. On the one hand, age, ex-pertise and concerns about infection risk increase the risk of suffering anxiety among frontline healthcare workers fighting COVID-19 [23], on the other hand, lack of PPE, and fear of infection increase the risk among nursing students [24], as well as a gender associ-ation: there is a higher anxiety level in female students than in males [2,25-28].
- The study phenomenon regarding CPR training wearing PPE of COVID should be described in the background yet.
Answer: In this point, we have added a paragraph considering recent literature.
Lines 74-78: Respect to the effect of wearing personal protective equipment on CPR quality, during the pandemic, Rauch et al. [33] have recently published the results from a sample of providers from the prehospital emergency medical service. In that study, they didn’t find any effect of wearing PPE with respect to compression depth, release, rate or number of effective compressions.
- The study design (cross-sectional) is not appropriate if the researchers conducted interventions.
Answer: Certainly, there was an error on the manuscript. On this reviewed one, we describe it as a before-after pilot study.
- How did the researcher calculate the sample size and randomize the participants?
Answer: 520 students were invited to participate because this is the total number of students in Nursing degree, but only 73 students were elegible participants due to “had no prior CPR (Cardiopulmonary Resuscitation) knowledge”, “did not wish to participate” or exclusion criteria (Figure 1). To avoid any bias, we designed a pilot study considering that we could have a small sample (maximum of 30 participantes) because the complexity of the materials used in and the simulated scenario.
Using random numbers generated by computer software XLSTAT ® BioMED 14.4.0 (Microsoft Inc., Redmond, USA) we appointed a final sample of 24 participants (Figure 1).
- Is it ethical to provide intervention without PPE? Did the researcher conduct ethical clearance? It is important since student is one of vulnerable subjects in research.
Answer: All the participants were informed about the general objectives of the study and gave their informed consent. As a pilot study in simulated scenarios, the participants make the CPR maneuvers with and without a PPE kit, simulating a biological risky situation, providing a realistic experience but without any real risk.
Besides, as we described in section 2.1 Study Design: “The study was approved by the Clinical Research Ethics Committee of Talavera de la Reina (Toledo) with number 178013/113. Details of the study design, statistical analysis plan, and baseline data are available online (doi.org/10.1186/ISRCTN10222040).”
- Please describe normality test result as the requirement to use t-test.
Answer: Thanking for the comment, we include that data were checked to meet the normality condition by the Shapiro Wilk test.
- Was there any qualitative study conducted in the study because the researchers asking about main concern about performing a CPR with PPE?
Answer: This study includes the use of an “ad hoc” survey for the study based on the model presented by Miguel Perez et al. [36] and Romo-Barrientos et al. [37]. Pre-feelings and emotions questionnaire consist of 12-questions and the post-feelings and emotions consists of 16-questions (with adding four new questions related to students’ satisfaction and emotional experience making the CPR) (Supplementary material).
This instrument was administered to all participants to characterize students’ feelings and emotions regarding the CPR maneuvers and previous use of the PPE. In these terms, we study a qualitative (self-view) aspect about CPR, asked by a survey, that have been analyzed as quantitative variables (see tables attached to the manuscript).
- Explanations about coping in the discussion are not consistent with results.
Answer: On the discussion section, we have tried to compare our outcomes with others, highlight the strengths and weaknesses of our study, and highlight the significance of the study. Thus, we mention that other results about coping strategies could be used to develop specific training in nursing students (line 282). Those are not our outcomes, however, the discuss how we could improve nursing clinical training considering our results about fears and attitudes, and considering other studies about anxiety, but also about coping strategies in healthcare students and professionals.
We would like to thank the referee again for taking the time to review our manuscript. Sincerely,
The authors

Reviewer 2 Report
First of all, thank you for your contribution to our journal.
It is judged that the purpose of our journal and this study's subject is incorrect, and it seems more appropriate to post in another journal.
If I may add a few words,
1. The study subjects were nursing degree students and nursing-podiatry degree students, but are they all nursing students? It's not a podiatrist course. I doubt it. In addition, only 520 to 48 subjects were selected, which has not been sufficiently considered for the selection and exclusion criteria. Regrettably, this is a small or improperly selected sample.
2. Abstract- The description of statistic analysis is missing.
3. Regarding research necessity, nurses said that most of them are women, so they study women and men separately. However, it is not appropriate to distinguish women and men for that purpose. If you insist on conducting research, it seems more appropriate to design a factor study on anxiety only for female nursing college students. And why are simulation classes wearing PPE in COVID-19 situations so you can learn decision-making, learning process, and coping skills?
There needs to be more explanation about the relationship between decision-making and learning. This lacks a description of the research basis and a sufficient description of the research background in the introduction.
4. The purpose of the research presented in the introduction is to experiment with attitudes, fears, and anxiety. However, the text only contains information about anxiety, so it does not fit the context.
5. There is no explanation for the research tool for attitude and fear
Anxiety tools also have no results of reliability validity verification in this study
6. The research design suggested that pre-post experiments were conducted, but in Figure 1, the study subjects were divided into groups wearing PPE and those not wearing PPE.
Also, statistical analysis methods are more appropriate than ANOVA.
Therefore, the consistency of research design and subject selection, and research analysis methods are unreliable and inappropriate in research design and statistical analysis methods.
6. Part of the research results - The table for statistical analysis results seems incomplete, and most of the p-values are higher than 0.10, which makes it clear that the researcher made a mistake in establishing the research hypothesis.
7. The focus is unclear whether the research design is performing CPR wearing PPE or measuring the degree of anxiety about simple CPR techniques.
8. The result is repeating the results without interpretation of the research results, and many suggestions cannot be made with the research data provided in the discovery part, so it seems unconvincing.
Author Response
Dear Editor,
We wish to thank you all for your constructive comments in this round of review. Your comments provided valuable insights to refine our contents and analysis.
Thus, the manuscript has been revised according to the suggestions and comments of the reviewers. Please also kindly note that the title of the manuscript has been changed to “Gender Differences in Anxiety, Attitudes and Fear, among Nursing Undergraduates Coping with CPR Training with PPE Kit for Covid”.
The responses to the specific comments of the reviewers are as follows:
From Reviewer 2:
- The study subjects were nursing degree students and nursing-podiatry degree students, but are they all nursing students? It's not a podiatrist course. I doubt it. In addition, only 520 to 48 subjects were selected, which has not been sufficiently considered for the selection and exclusion criteria. Regrettably, this is a small or improperly selected sample.
Answer: All the students are from the Nursing degree. We have reviewed this paragraph. 520 students were invited to participate because this is the total number of students in Nursing degree, but only 73 students were elegible participants due to “had no prior CPR (Cardiopulmonary Resuscitation) knowledge”, “did not wish to participate” or exclusion criteria (Figure 1). To avoid any bias, we designed a pilot study considering that we could have a small sample (maximum of 30 participantes) because the complexity of the materials used in and the simulated scenario.
Using random numbers generated by computer software XLSTAT ® BioMED 14.4.0 (Microsoft Inc., Redmond, USA) we appointed a final sample of 24 participants (Figure 1)
- Abstract- The description of statistic analysis is missing.
Answer: A statistical description has been added in the abstract: “We studied by t-test: STAI variables according to sex and the physiological values related to anxiety level of participants. ANOVA statistical test was used to perform the data analysis of STAI variables.”
- Regarding research necessity, nurses said that most of them are women, so they study women and men separately. However, it is not appropriate to distinguish women and men for that purpose. If you insist on conducting research, it seems more appropriate to design a factor study on anxiety only for female nursing college students. And why are simulation classes wearing PPE in COVID-19 situations so you can learn decision-making, learning process, and coping skills?
There needs to be more explanation about the relationship between decision-making and learning. This lacks a description of the research basis and a sufficient description of the research background in the introduction.
Answer: The study of anxiety (TA and SA) has been analyzed out globally (table 1 Total). Our sample has the same proportion between men and women (45.8% and 54.2% respectively). The gender perspective in this study could provide a view of gender differences; although there is a high percentage of female registered nurses.
We have considered that`s important to assess the anxiety levels of our students in clinical simulation practices, because high anxiety is directly related to learning difficulties, as we referred to on the manuscript. Many studies have shown that high level of anxiety can hinder the learning process, and anxiety situations can also affect making wrong decisions.
It is necessary to know the gender of the participants since the strategy to reduce anxiety is different between men and women since men do not present anxiety prior to the activity
The reason for PPE in COVID-19 situations is since this study reflects a current clinical scenario. In those terms, undergraduates have included practices with PPE in their curriculum, with is useful for any situation in what that specific level of biosecurity is needed.
Attending to the reviewer suggestion, we have added the sentence "training with PPE kit in order to enable them to work in the covid scenario or in others that require that level of biosecurity"
Finally, a paragraph considering the learning and decision-making process has been added to the introduction.
- The purpose of the research presented in the introduction is to experiment with attitudes, fears, and anxiety. However, the text only contains information about anxiety, so it does not fit the context.
Answer: Anxiety (state and trait anxiety) has been studied with the STAI questionnaire, a validated questionnaire in Spanish, a widely used tool for this type of research.
To evaluate the feelings and emotions, two questionnaires have been developed (pre and post, as detailed in the methodology), adapted from Miguel Perez et al 2007 and used by different authors (Leboulanger, 2011; Criado et al., 2017; Romo- Barrientos et al., 2019, 20, 22) and published in different journals such as: Annals of Anatomy, Anatomical Science Education, European Annals of Otofhinolaryngology, Head and Neck diseases. BMC Media Education.
The results of these pre-post questionnaires for feelings and emotions are collected in figures 2 and 3. We attach the questionnaires “feelings and emotions”.
- There is no explanation for the research tool for attitude and fear Anxiety tools also have no results of reliability validity verification in this study.
Answer: Thanking the comment, let us explain that aanxiety tool was the STAI questionnaire. It is validated for the Spanish population and has a Cronbach’s alpha of 0.93 for TA and 0.92 for SA (Spielberger 2019).
For feelings and emotions, two questionnaires have been developed (pre and post, as detailed in the methodology), adapted from Miguel Perez et al 2007 and used by different authors (Leboulanger, 2011; Criado et al., 2017; Romo- Barrientos et al., 2019, 20, 22) and published in different journals such as: Annals of Anatomy, Anatomical Science Education, European Annals of Otofhinolaryngology, Head and Neck diseases. BMC Media Education.
Both the STAI questionnaire and the two self-emotional questionnaires have been detailed in the methodology section. In addition, these last two questionnaires have been uploaded as complementary material in this reviewed version.
- The research design suggested that pre-post experiments were conducted, but in Figure 1, the study subjects were divided into groups wearing PPE and those not wearing PPE. Also, statistical analysis methods are more appropriate than ANOVA.
Answer: Certainly, we describe it as a before-after pilot study, but we compared it according to wearing PPE in first journal or not wearing PPE. This design was chosen because this study corresponds to a bigger project (https://www.isrctn.com/ISRCTN10222040).
Flow chart was wrong, so we have proceeded to modify figure 1 to reflect the real flowchart of the participants.
We studied by t-test: STAI variables according to sex and the physiological values related to anxiety level of participants and ANOVA statistical test was used to perform the data analysis of STAI variables (PRE-trait anxiety vs PRE-state anxiety vs POST-state anxiety) in table 1.
Therefore, the consistency of research design and subject selection, and research analysis methods are unreliable and inappropriate in research design and statistical analysis methods.
Answer: In this point, we agree that the reliability of the results of the study as a whole, can be enhanced internally by carefully monitoring and controlling issues that might contribute to inconsistency in design like (a) changes over time due to self-selection of participants into (we didn’t have those changes); (b) maturation in the participants (we didn’t have that problem), or (c) subject/researcher expectancy effects. To avoid any bias, we designed a pilot study considering that we could have a small sample because the complexity of the materials used in and the simulated scenario. Even it is a small sample, we could extrapolate our results for the nursing students collective in the region. Reviewers must know also that this is a pilot study that we would like to extend to a big and more representative sample.
- Part of the research results - The table for statistical analysis results seems incomplete, and most of the p-values are higher than 0.10, which makes it clear that the researcher made a mistake in establishing the research hypothesis.
Answer: Table 1 has been modified to better understand the data information.
Certainly, we have a small sample, with affect to the outcome’s significance; however, the study is carried out for a 95% confidence level.
The degree of significance is not directly related to the sample size even so, considering the reviewer suggestions, we must add this weakness in the study.
- The focus is unclear whether the research design is performing CPR wearing PPE or measuring the degree of anxiety about simple CPR techniques.
Answer: the aim of this study was to examine the attitudes, fears, and anxiety level, that nursing students experience when faced with a critical clinical simulation (cardiopulmonary reanimation) with and without a personal protective equipment. the objective has been more detailed in the abstract and introduction. The objective of the study is to study anxiety and feelings. We have proceeded to change the title.
- The result is repeating the results without interpretation of the research results, and many suggestions cannot be made with the research data provided in the discovery part, so it seems unconvincing.
Answer: At this point, we have made a description of the results in the results section; we do not have to make an interpretation of them in that section. Considering your comment, we have discussed the results with what is published in that topic.
We would like to thank the referee again for taking the time to review our manuscript. Sincerely,
The authors

Round 2
Reviewer 2 Report
Your manuscript looks much more organized and improved.
Thank you for your hard work.
We want the abstract to be based on facts about the research results. Therefore, please accurately describe, instead of 520 participants, 24 participants, how anxiety levels differed by gender. And be sure to correct any typos in the text. (e.g., PCR->cpr)
Author Response
Thank you for your comments.
We include your suggestions in the abstract.